# The Role of Primary Cilia-Associated Phosphoinositide Signaling in Development

**DOI:** 10.3390/jdb10040051

**Published:** 2022-12-02

**Authors:** Chuan Chen, Jinghua Hu, Kun Ling

**Affiliations:** 1Department of Biochemistry and Molecular Biology, Mayo Clinic, Rochester, MN 55905, USA; 2Division of Nephrology and Hypertension, Mayo Clinic, Rochester, MN 55905, USA

**Keywords:** phosphoinositide kinases and phosphatases, primary cilia, development

## Abstract

Primary cilia are microtube-based organelles that extend from the cell surface and function as biochemical and mechanical extracellular signal sensors. Primary cilia coordinate a series of signaling pathways during development. Cilia dysfunction leads to a pleiotropic group of developmental disorders, termed ciliopathy. Phosphoinositides (PIs), a group of signaling phospholipids, play a crucial role in development and tissue homeostasis by regulating membrane trafficking, cytoskeleton reorganization, and organelle identity. Accumulating evidence implicates the involvement of PI species in ciliary defects and ciliopathies. The abundance and localization of PIs in the cell are tightly regulated by the opposing actions of kinases and phosphatases, some of which are recently discovered in the context of primary cilia. Here, we review several cilium-associated PI kinases and phosphatases, including their localization along cilia, function in regulating the ciliary biology under normal conditions, as well as the connection of their disease-associated mutations with ciliopathies.

## 1. Introduction

Phosphoinositides (PIs) are reversibly phosphorylated derivatives of the membrane phosphatidylinositol (PtdIns, Figure 1) [1,2]. PIs consist of two fatty acid chains, a glycerol moiety, and a D-myo-inositol-1-phosphate head group that is decorated by phosphate moiety at the 3-, 4-, and/or 5-hydroxyl position to produce seven phosphorylated derivatives (PtdIns3P, PtdIns4P, PtdIns5P, PtdIns(3,4)P_2_, PtdIns(3,5)P_2_, PtdIns(4,5)P_2_, and PtdIns(3,4,5)P_3_) [1]. The exact composition of each species varies in different subcellular locations and different cell types. Compared to other phospholipids in cellular membrane compartments, PIs are relatively minor constituents and PtdIns4P and PtdIns(4,5)P_2_, which contribute the largest amounts, are less than 1% of the total cellular phospholipid pool [1]. Despite the low abundance, PIs are indispensable to the integrity of cells. PIs play essential roles in various cellular events including, but not limited to, cytoskeletal dynamics, cell adhesion and migration, vesicular trafficking, assembly of cargo proteins at membranes, cell proliferation, and survival [1,2,3,4]. Importantly, PIs are key identity determinants of various cellular membrane compartments and act as spatiotemporal cues to direct signaling events [2,3,5,6,7].

Each PI species exhibits unique distribution across the different plasma membrane (PM) and organelle membrane compartments [2]. In general, the lipid tail of PIs inserts into membrane compartments, whereas the phosphorylated inositol head of individual PIs recognizes and binds to unique PI effector proteins [2,8,9]. Current reported PI-binding domains include the Pleckstrin Homology (PH), FYVE, Phox (PX), C2, PROPPIN, PTB, Tubby, TRAF, ANTH/ENTH, FERM, and PDZ domains [9,10,11]. The affinity and specificity vary between different PIs and binding domains [9]. Individual PI species induce the conformational change of effectors and/or recruits them to specific subcellular locales to meet their signaling partners [8,9,12,13]. This leads to the activation/inhibition of PI effectors at unique subcellular locales [8]. Thus, the precisely controlled spatiotemporal availability of PI species in the cell is critical. PIs metabolism and homeostasis are maintained and regulated by specific PI kinases and phosphatases that are conserved in evolution [3,5,8]. Many reports have discovered that the dysfunction of components of the PI signaling pathway is responsible for various human diseases, from rare genetic disorders to the most common conditions such as cancer, obesity, neurological disorders, cardiovascular disease, and diabetes [7,14,15,16,17,18].

The primary cilium is a sensory organelle presenting on the surface of most mammalian cell types [19]. Although the ciliary membrane is reported to be continuous with the plasma membrane, the lipid and protein composition in the ciliary membrane is maintained unique and distinguishable from the plasma membrane [20,21]. To fulfill its function as a signaling hub and cellular antenna, the trafficking of protein and membrane components in and out of cilia is highly active and tightly controlled, suggesting the potential involvement of PIs in cilia. Indeed, it was reported in 2009 that mutations in an inositol polyphosphate 5-phosphatase, INPP5E, are causal for multiple ciliopathies [22,23]. After Subsequently, more studies implicated the connection between PIs and primary cilia. PIs are reported to regulate the stability/disassembly of primary cilia [24,25], ciliogenesis [26], and ciliary trafficking [27,28]. The dysfunction of multiple PI enzymes, both phosphatases, and kinases, is connected to ciliary defects and ciliopathies [6]. The localization and function of PIs in cilia have been reviewed in several excellent recent reviews [6,29,30]. Here, we mainly focus on the role of several cilia-associated components of the PI signaling pathway in development, tissue homeostasis, and related diseases.

## 2. Primary Cilia

### 2.1. Structure of Primary Cilia

The primary cilium is a sensory organelle presenting on the surface of most quiescent cells in vertebrates [19,31]. The primary cilium is composed of a microtubule-based core structure called the axoneme, which is surrounded by a ciliary membrane that is continuously extended from the plasma membrane (Figure 2) [32]. Different from motile cilia, which can generate a fluid flow on cells such as respiratory epithelial and ependymal cells, primary cilia only contain nine microtubule doublets without two centrally located singlet microtubules (Figure 2) and are non-motile for lacking the axonemal dynein arms on the outer microtubules [33].

The basal body, transformed from the mother centriole upon serum starvation, nucleates the outgrowth of the axoneme and forms the base of primary cilia. The region between the axoneme and basal body is identified as the transition zone, which is characterized by Y-links connecting the axonemal microtubules to the ciliary membrane [34]. Transition fibers, which are transformed from distal appendages of the mother centriole, are located below the transition zone on the distal end of the basal body [35]. In addition to anchoring the basal body to the plasma membrane, transition fibers function together with the transition zone as the ciliary gate to regulate ciliogenesis and signaling by controlling the selective transport of specific ciliary proteins between the cell body and the cilium [35,36]. Intriguingly, many transition zone and transition fiber proteins are reported to contain phosphoinositide binding domains; however, the physiological significance of their PI binding potential remains unclear [37].

### 2.2. Ciliopathy

Although largely considered a vestige before 1999, the primary cilium is now demonstrated as a signaling center to interpret both chemical and mechanical signals. Major components of multiple signaling pathways specifically localize in primary cilia [38]. These include pathways that are essential for the biogenesis and homeostasis of tissues and organs, such as the Hedgehog signaling [19], canonical Wnt signaling [39], Planar cell polarity [40], G protein-coupled receptor signaling [41], as well as tyrosine kinase receptor signaling [42] pathways. The activation and regulation of these pathways occur in cilia and depend on the normal structure and functionality of primary cilia [19].

Consistently, defects in primary cilia lead to a panel of genetic human diseases, collectively termed ciliopathies, including numerous seemingly unrelated syndromes, with involvement of the brain, eye, kidney, liver, pancreas, skeletal system, and some others in a growing list [43,44,45,46,47,48]. Common ciliopathies range from organ-specific disorders such as autosomal dominant polycystic kidney disease (ADPKD) and autosomal recessive PKD (ARPKD) to pleiotropic disorders such as Joubert syndrome (JBST), nephronophthisis (NPHP), Bardet–Biedl syndrome (BBS), and Meckel syndrome (MKS) [49,50]. Currently, more than 190 established and 250 candidate cilium-associated proteins are identified as ciliopathy proteins and over 35 ciliopathies are reported, emphasizing the significance of primary cilia in development and human disease [51,52]. Given the physiological significance of the primary cilium, the molecular mechanisms underlying its biogenesis, function, and related disease pathogenesis have just started to be explored in the recent two decades.

## 3. Cilium-Associated Phosphoinositides and PI-Binding Ciliary Proteins

### 3.1. Cilium-Associated Phosphoinositides

Although continuously extended from the PM, the ciliary membrane possesses unique composition and compartmentalization of PIs (Figure 3). PtdIns4P and PtdIns(4,5)P_2_ are the most well-studied ciliary PIs. PtdIns4P is the dominant phosphoinositide along the ciliary membrane, whereas PtdIns(4,5)P_2_ is mainly limited to the proximal part of the cilium and the ciliary base [28,53]. Different from the Golgi membrane that enriches PtdIns4P by residential PI4Ks, this unique PtdIns4P compartmentalization in the ciliary membrane is achieved and maintained by the ciliary PI 5-phosphatase INPP5E [27,28]. Loss of INPP5E from cilia leads to the replacement of ciliary PtdIns4P by PtdIns(4,5)P_2_. Increased ciliary PtdIns(4,5)P_2_ disrupts Hedgehog signaling through recruitment of the PtdIns(4,5)P_2_-binding, trafficking protein Tubby-like protein-3 (TULP3) and its cargoGPR161, the Hedgehog negative regulator G protein-coupled receptor, into cilia [26,27,28]. Additionally, PtdIns(4,5)P_2_ regulates the stability of primary cilia by balancing membrane turnovers. Ciliary remodeling PtdIns(4,5)P_2_ is reported to cause ciliary disassembly through ciliary vesicle release by actin polymerization [54,55,56]. This process is likely to be modulated by ciliary disassembly regulator Aurora A Kinase (AURKA) with involvement of Histone Deacetylase 6 (HDAC6)-dependent microtubule disassembly. However, contemporary studies only show that both inhibitors of AURKA and HDAC6 ameliorate the ciliary disassembly induced by acute PtdIns(4,5)P_2_ synthesis [56,57], leaving the underlying molecular mechanism unexamined.

PtdIns(4,5)P_2_ and PtdIns(3,4,5)P_3_ are also reported to accumulate at the transition zone in the *Inpp5e*-null mouse embryonic fibroblast cells (MEFs) upon Hedgehog signaling activation and are associated with the reduced ciliary accumulation of Smoothened (SMO) and GLI2 [24], two important components of the Hedgehog pathway. INPP5E by dephosphorylating PtdIns(3,4,5)P_3_ generates a PtdIns(3,4)P_2_ pool at the basal body, which together with PtdIns(3,4,5)P_3_ activates PI3K/AKT and helps maintain the ciliary stability [24,25]. Moreover, a PtdIns3P pool is discovered in the pericentriolar recycling endocytic compartment around the base of primary cilia, which is likely maintained by a class II PI 3-kinase PI3K-C2α enriched at the ciliary base [58]. This specific pool of PtdIns3P is necessary for the activation of Rab8 and Rab11, which is required for the axoneme elongation and the ciliary transport of polycystin-2 and Hedgehog signaling components [58,59,60].

### 3.2. PI-Binding Ciliary Proteins

It has been well documented that PIs are important regulators of membrane trafficking, protein transportation, and protein complex assembly in various subcellular locations [4,5,21], which are key events for the ciliary signaling. Similarly to in the cytoplasm, PIs exhibit different functions in cilia in different subdomains, and this spatiotemporal availability/functionality of specific PI species is achieved by the specific ciliary localization of relevant phosphatases, kinases, as well as PI-binding effectors. Among all the reported PI-binding domains, PH, C2, Tubby and B9 domains are found in primary cilium-associated proteins [28,37,61,62,63].

PH domains are well-characterized signaling modules identified in diverse proteins [64]. A small subset of PH domains (about 10–20%) bind individual PIs, commonly PtdIns(3,4,5)P_3_, PtdIns(3,4)P_2_, PtdIns4P, and PtdIns(4,5)P_2_ [64,65]. As one of the most well-studied PI-effectors, AKT has an N-terminal PH domain which, by binding to PtdIns(3,4,5)P_3_, recruits AKT to the plasma membrane in response to growth factor stimulation, which is required for the consequent phosphorylation and activation of AKT [62]. The phosphorylated and activated AKT is also reported to localize at the ciliary base and facilitate the ciliogenesis by phosphorylating the ciliary structural protein Inversin (INVS), which is a causative gene for ciliopathy nephronophthisis type II (NPHP2) [53,61,66]. In Hedgehog-dependent medulloblastoma cells, a compartmentalized PtdIns(3,4,5)P_3_/AKT/GSK3β signaling axis is also identified at the ciliary base, where AKT phosphorylates and inhibits GSK3β, leading to cilia loss [25,60].

TULP3 is a PtdIns(4,5)P_2_-binding protein with a conserved C-terminal Tubby domain and a conserved N-terminal helix that interacts with the intraflagellar transport complex IFT-A [67]. TULP3 is required for the ciliary localization of multiple membrane-associated proteins [67,68,69,70]. GPR161 is a Gαs-coupled receptor that suppresses the Hedgehog pathway by increasing cAMP and PKA activity [68,71]. In *Inpp5e* KO cells, accumulated PtdIns(4,5)P_2_ in cilia attracts more TULP3 and subsequently more GPR161 in cilia for the repression of the Hedgehog signaling [27,28]. Reducing TULP3 levels restored the activation of Hedgehog pathway in *Inpp5e* KO cells, supporting that TULP3 is the key effector influenced by the availability of INPP5E and PtdIns4P [28]. On the other hand, TULP3 is also necessary for the ciliary localization of INPP5E, as well as other proteins bound to the ciliary membrane including the small G-protein ARL13B and ADPKD disease proteins polycystin 1 and polycystin 2 [69,70]. Current studies suggested that the IFT-A-binding motif instead of the PtdIns(4,5)P_2_-binding domain of TULP3 is necessary for the ciliary localization of those membrane proteins in normal conditions [70].

Many transition zone proteins contain C2 domains, such as AHI1, NPHP1, NPHP4, MKS5/RPGRIP1L/NPHP8 and CC2D2A/MKS6 [37,72], which have the potential to bind PIs. However, whether they indeed bind to PIs and/or regulated by PI binding have not been well characterized. MKS-5 acts as an assembly factor for the establishment of the transition zone [73,74]. In *Caenorhabditis elegans*, it was proposed that the C2 domains of MKS-5 interact with other transition zone proteins, which potentially forms a lipid gate to limit the abundance of PtdIns(4,5)P_2_ within cilia [73]. This study showed that PtdIns(4,5)P_2_ diffused from the transition zone to the ciliary proper in the *mks-5* mutant [73]. The homeostasis of compartmentalized PtdIns(4,5)P_2_ is critical for the stability and signaling of cilia [28,55]; thus, it is worth investigating whether and how PtdIns(4,5)P_2_ localization is controlled by this MKS5-centered lipid barrier in vertebrates. B9 domains are specific ciliary C2 domains that are identified in MKS1, B9D1/MKS9, and B9D2/MKS10 [37,72,75]. These transition zone proteins form a linear MKS1–B9D2–B9D1 tripartite complex through the binding of B9 domain and are required for the diffusion-barrier function of the transition zone [63,76]. Similarly, the phosphoinositide binding property of these B9 domains is unclear.

## 4. Primary Cilium-Associated PI Phosphatases

### 4.1. INPP5E

INPP5E dephosphorylates the 5-position on the inositol ring of PtdIns(4,5)P_2_ and PtdIns(3,4,5)P_3_ and converts them to PtdIns4P and PtdIns(3,4)P_2_, respectively [1]. In non-ciliated cells, INPP5E resides at distal appendages of the mother centriole and maintains a specific centrosomal PtdIns4P pool [26]. PtdIns4P binds to both tau-tubulin kinase-2 (TTBK2) and the distal appendage protein CEP164, which compromises the TTBK2-CEP164 interaction and inhibits the recruitment of TTBK2 [26]. In quiescent cells, INPP5E translocates to cilia with a decreased PtdIns4P level at centrosome, leading to the recruitment of TTBK2 and the initiation of ciliogenesis [26]. During ciliogenesis, INPP5E translocates to the ciliary membrane, where it maintains the stability and signaling function of primary cilia by preserving a PtdIns4P-dominant environment (Figure 3) [23,26]. INPP5E contains an N-terminal proline-rich domain, an inositol polyphosphate 5-phosphatase domain, and a C-terminal CAAX prenylation motif which is necessary for its ciliary localization [1,23]. As reported, phosphodiesterase PDE6D, the transition fiber protein CEP164, and the small GTPase ARL13B coordinate to recruit INPP5E into the ciliary membrane [77,78,79].

Current knowledge from human patients and animal models supports that INPP5E is essential for development. Two human ciliopathy syndromes are associated with *INPP5E* mutations: Joubert syndrome (JBTS) and MORM syndrome (Table 1 and Table 2) [80,81,82,83]. JBTS is an autosomal recessive neurodevelopmental disorder characterized by the appearance of a “molar tooth sign” on axial MRI, which results from the abnormal development of the cerebellar vermis and the brainstem [84]. The most common clinical features of JBTS include ataxia, hyperpnea, sleep apnea, ocular motor apraxia, hypotonia, and cystic dysplastic kidney (Figure 4 and Table 2) [85]. JBTS-associated INPP5E mutations are mainly missense and cluster in the catalytic domain (Table 1). In vitro experiments also confirmed that these muted INPP5E exhibit decreased phosphatase activity and cause impaired PI distribution in cilia [81,83,85,86,87]. On the other hand, MORM syndrome is resulted from C-terminal deletions of INPP5E (Table 1), which leads to loss of the CAAX motif and exclusion of INPP5E from the ciliary membrane without affecting the catalytic activity [23,88]. Correspondingly, in addition to similar symptoms as JBTS including intellectual disability and retinal dystrophy, MORM syndrome shows phenotypes unseen in JBTS, including obesity and micropenis, as observed in other ciliopathies such as Bardet–Biedl syndrome and Cohen syndrome (Figure 4 and Table 2) [23,88,89]. Compared with JBTS-associated INPP5E mutants which lead to a loss of INPP5E activity in the whole cell and whole body, MORM-associated INPP5E mutants only damage the INPP5E activity in cilia but may increase the INPP5E activity in other cellular locales due to mislocalization. In this context, the phenotypic differences between JBTS and MORM may be caused by the abnormal non-ciliary activity of INPP5E, which could be seen when other ciliary abnormalities disturb the ciliary targeting of INPP5E.

Results from animal models reinforce the conclusion in human patients that INPP5E function in primary cilia is required for the development of multiple organs [23,24,90]. *Inpp5e*^D/D^ mice (deletion of exons 7 and 8) are embryonic and postnatal lethal with ciliopathy features, including bilateral anophthalmos, postaxial hexadactyly, renal cyst, skeletal abnormalities, as well as cerebral developmental defects, suggesting that INPP5E is essential for primary cilium-mediated functions [23]. Interestingly, the inactivation of *Inpp5e* in mice on the postnatal day-28 did not affect the survival of adult mice; however, still caused ciliopathy phenotypes such as higher body weight, retinal dystrophy, and cystic glomeruli [23]. This result supports the involvement of INPP5E in the cilium-dependent homeostasis of mature tissues/organs, whereas also indicates that the INPP5E-dependent ciliary function is more important for the embryonic development. Further studies using the *Inpp5e*^D/D^ MEFs suggest that the *Inpp5e* deletion increases the ciliary level of the Hedgehog suppressor GPR161 via PtdIns(4,5)P_2_-dependent recruitment of TULP3, which promotes GLI3R formation and reduces Hedgehog signaling [28]. Consistent with the *Inpp5e*^D/D^ mice, another mouse model of *Inpp5e^-/-^* (deletion of exons 2 to 6) also recapitulates JBTS features, including polycystic kidneys, cleft palate, polydactyly, edema, and ossification delays [23,24,90]. *Inpp5e^-/-^* MEFs exhibited the defective Hedgehog signaling with reduced ciliary accumulation of SMO and GLI2 when treated with the SMO agonist [24]. Expression of a constitutively active SMO mutant (SMOM2) in *Inpp5e^-/-^* mice partially restored the associated embryonic development defects, emphasizing the involvement of Hedgehog signaling in INPP5E regulation in embryonic development [24]. However, whether the expression of SMOM2 can rescue the defects in *Inpp5e^-/-^* adult mice is not determined in this study.

Although most INPP5E-associated JBTS patients mainly show neurologic symptoms with rare kidney or hepatic features, both *Inpp5e*^D/D^ and *Inpp5e^-/-^* mice showed polycystic kidneys with a 100% penetrance [23,24]. Moreover, the renal-specific deletion of *Inpp5e* exons 2-6 in mice resulted in severe polycystic kidneys and renal failure, likely caused by the hyperactivation of PI3K/Akt and mTORC1 pathway [90]. Despite the possibility that INPP5E may carry slightly diverse functions in different species, that the PKD phenotype in *Inpp5e*-inactivated mice is absent in *INPP5E*-mutated human patients more likely reflects the dosage difference of functional INPP5E in both models. Mouse models suffer from a complete loss of INPP5E protein during the embryonic and/or postnatal development of kidneys [23,90], whereas in patients, INPP5E mutations are often hypomorphic with reduced protein level, weakened phosphatase activity, or defective ciliary localization [23,83], which may be sufficient to support the development and homeostasis of kidneys. In other animal model such as zebrafish, *Inpp5e* knockdown in morphants also impairs cilia formation and function in the Kupffer’s vesicle and pronephric ducts, thus leads to ciliopathy-like phenotypes including body axis asymmetry, microphthalmia, pericardial edema, kinked tail, and pronephric cyst formation [91]. Moreover, expression of human INPP5E rescues the defects in the *Inpp5e* knockdown morphants [91], suggesting that the functionality of INPP5E is mostly conserved in vertebrate development.

Especially, INPP5E patient mutations exhibit reduced cilia stability compared with the wild-type protein [23]. Taken together, INPP5E is the most well-studied PI enzyme, which plays a critical role in cilia assembly, stability, and signaling pathways [6,29,92,93], highlighting the importance of INPP5E in development.

### 4.2. OCRL and INPP5B

OCRL and INPP5B are both members of the inositol polyphosphate 5-phosphatase family as INPP5E, but mainly hydrolyze PtdIns(4,5)P_2_ to generate PtdIns4P [1]. OCRL localizes to the cilium and basal body, as well as the endocytic network, and functions in the assembly and maintenance of primary cilia (Figure 3) [94,95,96,97,98]. OCRL is a multi-domain protein including PH, 5-phosphatase, ASH (ASPM-SPD2-Hydin), and catalytically inactive RhoGAP (Rho GTPase-activating protein) domains [1]. At the early stage of ciliogenesis, OCRL is recruited by the small GTPase RAB8 to cilia through direct binding [95,99], which is essential for primary cilia assembly [94]. Mutations of OCRL in the 5-phosphatase domain interrupts its ciliary localization, whereas deletion of the RhoGAP domain eliminates OCRL in the ciliary proper and restrains it near the ciliary base [95]. This is consistent with the discovery that patients’ fibroblasts with *OCRL* mutations exhibit defective ciliogenesis and shortened cilia [95,96], indicating the diverse disorders present in *OCRL*-mutated patients may be due to the dysregulation of primary cilia. As a paralog of OCRL, INPP5B shares similar structural domains [1] but contains a C-terminal CAAX prenylation domain, which is essential for the ciliary localization of INPP5B [100]. Knockdown of *INPP5B* results in a significant decrease of ciliogenesis and ciliary length in both cultured mammalian cells and zebrafish Kupffer’s vesicle, but the detailed mechanism remains unclear [100].

*OCRL* mutations are identified as causative of two human diseases: oculocerebrorenal syndrome of Lowe (OCRL), also called Lowe syndrome, and Dent-2 disease (Table 1 and Table 2) [101,102,103,104,105]. Both Lowe syndrome and Dent-2 disease are rare X-linked genetic disorders. Lowe syndrome patients exhibit defective cilium assembly and ciliopathy symptoms such as intellectual disability, congenital cataracts, and renal dysfunction (Figure 4) [105]. Dent-2 is often described as a milder form of Lowe syndrome, as most patients only develop renal symptoms and a few have mild intellectual disability, hypotonia, cataracts, and rickets (Figure 4) [101]. Genetic analyses showed that Lowe syndrome-associated mutations are mostly in exons 8–23 (5-phosphatase, ASH, and RhoGAP domains), whereas Dent-2-associated mutations are typically in exons 1-7 (PH domain) (Table 1) [101,104], suggesting that the distinct symptoms in these two disorders result from different mutations. Interestingly, one study using different Lowe and Den-2 fibroblasts displays similar reduced protein and accumulated PtdIns(4,5)P_2_ levels but milder ciliogenesis defects in Den-2 [96], indicating the ciliation defect may be related to the disease severity. Moreover, the epigenetic differences among individual patients and/or secondary genetic mutations should be considered because the disease severity varies widely between patients who carry the same *OCRL* mutation [105,106]. However, INPP5B has not been associated with any human diseases. Thus, although INPP5B’s function may be partially redundant with OCRL [107], OCRL is the dominant enzyme accountable for corresponding developmental needs in human.

Results from animal studies suggest that mice respond differently from humans to the loss of OCRL [107,108]. *Ocrl* knockout mice are fertile with normal kidneys, eyes, and brains, failing to recapitulate the phenotypes of Lowe syndrome [107]. However, *Ocrl;Inpp5b* double knockout mice are embryonic lethal [107] and the kidney-tubule-specific deletion of *Inpp5b* in *Ocrl^−/−^* mice phenocopy the tubulopathy disorder of Lowe Syndrome/Dent-2 [109]. These results indicate that *Ocrl* and *Inpp5b* carry redundant functions in mice, and normal mouse development can be conducted if the combined enzyme activity of *Ocrl* and *Inpp5b* is above certain threshold. Once levels of *Ocrl* and *Inpp5b* drop below this threshold, the severity of developmental defects negatively correlates with the remaining dosage of functional *Ocrl* and *Inpp5b*. Interestingly, knock-in of human *INPP5B* in *Ocrl;Inpp5b* mice corrected the lethality, but animals still showed phenotypes such as Lowe syndrome/Dent-2 disease including reduced postnatal growth, low molecular weight proteinuria, and aminoaciduria [108]. This is consistent with the previous discovery that although *INPP5B* and *Inpp5b* are highly conserved in most exons, the significant differences in exons 7 and 8 lead to different gene transcription, mRNA splicing, and primary protein sequence of between human and mouse [110]. Moreover, one study showed that Inpp5b expression level was dramatically higher in mouse trabecular meshwork cells than the same human cells [100]. This distinct expression may partly explain why *Inpp5b* and *INPP5B* compensate the loss of *Ocrl* in mice differently.

Although several studies have confirmed the ciliary localization of OCRL and INPP5B [94,95,100], the investigation of OCRL and INPP5B in cilia has just begun. OCRL can be recruited to cilia by RAB8, which is essential for ciliary assembly, and regulate ciliary protein trafficking in an Rab8- and endosome-dependent manner [94,95]. Knockdown of *Ocrl* or *Inpp5b* in zebrafish resulted in defective cilium formation in Kupffer’s vesicle and ciliopathy-like phenotypes including microphthalmia, body-axis asymmetry, microlens, distorted retinas, and hydrocephalus [95,100]. Double knockdown of both *Ocrl* and *Inpp5b* in zebrafish showed synergistic effects, suggesting that these two ciliary PI phosphatases may play some non-redundant function in zebrafish development [100]. Consistent with the observation in zebrafish, the lack of OCRL in human retinal pigmented epithelial cells and patients’ fibroblasts results in defective ciliogenesis and shortened cilia [94,95]. However, one study using MDCK cells demonstrated an increased ciliary length when knocking down OCRL [98], indicating a potential cell-specific function of OCRL. Further investigation on the molecular mechanism underlying these differences may help the understanding of the tissue-specific manifestations in Lowe syndrome/Dent-2 disease. As PI phosphatases, how OCRL and INPP5B regulate the disruption of ciliary PIs is also unclear. One study using Lowe syndrome patients’ fibroblasts and *Ocrl-null* MEFs shows increased PtdIns(4,5)P_2_ and decreased PtdIns4P in cilia, which is similar to the observation in *Inpp5e-null* MEFs [97]. Intriguingly, OCRL is also necessary for the activation of Hedgehog signaling, but through a different mechanism from INPP5E. Unlike INPP5E that suppresses the ciliary entry of GPR161 but not SMO [27,28], OCRL deficiency has no effect on GPR161, but disrupts the ciliary translocalization of SMO upon SAG treatment [97]. The underlying molecular mechanism and involvement of corresponding PI species is highly interesting and should be determined in future studies.

## 5. Primary Cilium-Associated PI Kinases

### 5.1. PI3K-C2α

PI3K-C2α is a phosphoinositide 3-kinase and phosphorylates PtdIns and PtdIns4P to produce PtdIns3P and PtdIns(3,4)P_2_, respectively (Figure 1) [1]. PI3K-C2α is involved in a wide range of biological events such as endocytosis [111,112,113], exocytosis [114], mitosis [115], and autophagy [116,117]. Recently, a specific accumulation of PI3K-C2α is observed in the pericentriolar recycling endocytic compartment (PC-REC) at the ciliary base (Figure 1), where it produces a local pool of PtdIns3P [58]. In PI3K-C2α-depleted cells, the reduction in pericentriolar PtdIns3P altered the organization of PC-REC around the ciliary base, reduced RAB11 activation and impaired the ciliary targeting of RAB8 [58]. By this means, PI3K-C2α regulates the elongation of cilia and the trafficking of ciliary proteins such as polycystin 2 in a RAB8-dependent manner [58,59,118].

Homozygous global knockout of *Pik3c2a* (deletion of exon 1) in mice leads to embryonic lethality between E10.5 and E11.5 with abnormalities in multiple organs [119]. In addition to the defective angiogenesis and vascular maturation observed in *Pik3c2a^-/-^* embryos [119], some developmental disorders typically detected in mice with deficient Hedgehog signaling were identified, such as disrupted cardiac tube looping and impaired left–right patterning [58,120,121]. Consistent with the observation, *Pik3c2a^-/-^* MEFs and embryos exhibit impaired cilium elongation and ciliary reduction in the Hedgehog signal transducer SMO [58]. Furthermore, the loss of one *Pik3c2a* allele worsened the renal cystic burden in two PKD mouse models (*Pkd2^+/-^* and *Pkd1^+/-^*) [59]. In vitro studies using MEFs and IMCD3 cells also confirmed the impaired ciliary targeting of polycystin-2, encoded by *Pkd2*, in PI3K-C2α-depleted cells [59], supporting an essential role of PI3K-C2α in cilia in mouse development.

Recently, a novel human syndrome with short stature, cataracts, secondary glaucoma, and skeletal malformations has been identified in homozygous loss-of-function mutations in *PIK3C2A* (which encodes PI3K-C2α) (Table 3) [122]. This syndrome shares many similar features to classical ciliopathy syndromes and is especially close to the Lowe syndrome caused by mutations in the PI 5-phosphatase *OCRL*, as discussed above [122]. Similar to the observation in mice models, patients’ fibroblasts exhibited decreased PtdIns3P and RAB11 levels at the ciliary base, as well as reduced cilia length, suggesting that *PIK3C2A* is a candidate ciliopathy gene [122]. Further identification of additional patients will improve our understanding of the genotype–phenotype correlation associated with *PIK3C2A* mutations.

Although PI3K-C2α can generate both PtdIns3P and PtdIns(3,4)P_2_, current evidence suggests that PI3K-C2α regulates cilia by producing PtdIns3P at the PC-REC [58,59]. It was proposed that PtdIns3P activates the regional RAB11/RAB8 cascade and regulates the trafficking of membrane proteins such as SMO and polycystin 2 into cilia [58,59]; however, the underlying molecular mechanisms remain to be illustrated. Nevertheless, the ciliopathy phenotypes observed in mouse models and human patients endorse the importance of PI3K-C2α in the context of primary cilia and cilia-dependent developmental events [58,59,122].

### 5.2. PIPKIγ

Type Iγ phosphatidylinositol-4-phosphate 5-kinase (PIPKIγ) is the main kinase generating PtdIns(4,5)P_2_ by phosphorylating PtdIns4P (Figure 1) [1]. PIPKIγ is encoded by *PIP5K1C*. Currently, six alternative splicing isoforms of PIPKIγ have been identified that differ from each other solely by the C-terminal extension sequences [123]. Only isoform 3 with a unique motif at the C-terminus targets the proximal ends of centrioles or the basal body (Figure 3) [123]. While suppressing the centriole amplification in proliferating cells, PIPKIγ in non-duplicating cells regulates ciliogenesis in a kinase-dependent manner [26,124]. In proliferating cells, a centrosomal PtdIns4P pool generated by INPP5E inhibits the recruitment of tau tubule kinase 2 (TTBK2) and the subsequent removal of microtubule capping protein CP110 from the distal end of the mother centriole, thus blocks the initiation of axoneme assembly [26]. Signals promoting ciliogenesis triggers the removal of INPP5E from the mother centriole, leaving the centrosomal PtdIns4P pool to be exhausted by PIPKIγ, promoting the TTBK2 recruitment and the downstream axoneme elongation [26,124].

Although PIPKIγ appears indispensable for ciliogenesis, it also plays essential role in a wide range of cellular events at various subcellular locales such as focal adhesion assembly and endocytic trafficking, likely via different splicing variants [123,125,126,127,128]. Up to date, the only reported human developmental disorder associated with PIPKIγ is the lethal congenital contractural syndrome type 3 (LCCS3), a severe form of arthrogryposis [129]. Caused by a single homozygous mutation in *PIP5K1C* (PIPKIγ p.D253N, a kinase-dead mutation), LCCS3 patients exhibit multiple joint contractures, severe muscle wasting, and atrophy (Table 3) [129], sharing similar features observed in LCCS10, which is in the same phenotypic series and defined as a ciliopathy [130]. However, the LCCS3-associated *PIP5K1C* mutation affects all splicing isoforms of PIPKIγ. Whether and to what extent the cilia-associated PIPKIγ isoform 3 contributes to the pathogenesis of LCCS3 remain to be determined. Future studies in the context of cilia using patient-derived fibroblast or epithelial cells may provide useful evidence to answer these questions. Similar to LCCS3 patients, *Pip5k1c*-interrupted mice are embryonic or postnatal lethal with severe developmental defects. Among three different *Pip5k1c*-KO mouse models [131,132,133], *Pip5k1c* knockout mice with a fusion of β-Gal to the first 32 amino acids of PIPKIγ were embryonic lethal at E10.5 and exhibited ciliopathy-like phenotypes including exencephaly and pericardial effusion [131]. However, the other two *Pip5k1c* KO mice models with deletion of exons 2–6 or deletion of exon 18 died within 24 h after birth, with no apparent ciliopathy-like phenotypes observed in this time frame [132,133]. This is not surprising because all three *Pip5k1c* editing approaches affect all PIPKIγ equally. To understand the cilium-dependent functionality of PIPKIγ in development, it is necessary to generate mouse models in which individual PIPKIγ isoform is inactivated in specific ciliated tissues/organs. In addition, we recently showed that hydrocephalus syndrome protein 1 (HYLS1), a ciliopathy protein, functions as a PIPKIγ activator at the ciliary base. In cultured renal epithelial cells, both PIPKIγ and HYLS1 are necessary for the assembly of NPHP module at the transition zone and promote the removal of ciliary GPR161 and activation of the Hedgehog pathway [124]. Surprisingly, the kinase activity of PIPKIγ is vital for HYLS1 to regulate the axoneme elongation but appears dispensable for the ciliary trafficking of Hedgehog signaling components, indicating that the PIPKIγ-HYLS1 axis plays multiple roles in the context of primary cilia [124].

## 6. Unanswered Questions and Perspectives

Comparing to what has been known for PIs in the plasma membrane and various cytosolic membrane compartments, the role of each PI species in the context of primary cilia just started to emerge. In addition to the PI species, PI kinases and phosphatases, and PI effectors we discussed above, recent studies revealed more PI metabolic enzymes that may function in the context of primary cilia [134,135,136]. For example, the PI 3-phosphatase and tensin homolog (PTEN) (Figure 1) [1], which is mainly identified as a potent tumor suppressor [137], also functions significantly in multicilia formation and cilia disassembly by controlling the phosphorylation of the WNT signaling component Dishevelled in *Xenopus* [134]. Loss of phosphatidylinositol 4-kinase β (PI4KB), the main generator of PtdIns4P in the Golgi membrane (Figure 1) [1], in zebrafish led to the absence of primary cilia and ciliopathy phenotypes including neuromasts, pronephric ducts, and edema [135]. Moreover, the Src homology-2-domain-containing inositol-5-phosphatase SHIP2 was reported to increase the ciliary length and stability by confining AURKA at the basolateral membrane in polarized MDCK cells that formed spherical acini/cysts in 3D Matrigel, which is important for the lumen formation during the morphogenesis of epithelial tubules [136]. Future studies are needed to determine the mechanistic connection of these PI enzymes with the primary cilia and cilia-dependent cellular and biological processes that influence the development and tissue/organ homeostasis.

PIs are not only essential signaling components in various cell membranes, but also critical structural components to retain specific PI-binding proteins and determine the identity of these membrane domains/compartments [2,3,5]. Due to the interconvertibility, PIs together with PI kinases and phosphatases at the interface between different membrane compartments are essential integrators of membrane dynamics and facilitate the exchange/trafficking of proteins between membrane compartments [5]. In the context of primary cilia, PIs should act following similar mechanistic principles, which is proven true for the dynamics of PtdIns4P and PtdIns(4,5)P_2_ in the context of cilia [27,28]. However, systemic studies are needed to understand whether, which, and how PIs contribute to the transport of proteins and lipids from the endosomal membrane compartments to pericentriolar vesicles surrounding the basal body, through the ciliary gate, in the ciliary membrane to the ciliary tip, and then back to the ciliary base for recycling or activating downstream signaling molecules. To this end, determination of the precise localization and dynamics of PIs, as well as PI enzymes and effectors, at specific ciliary subdomains is critical. Although PI biosensors and antibodies were successfully employed to visualize various PI species and PI-binding proteins in cytoplasmic membranes, their application in cilia is harder due to the low abundance of PIs, the highly selective ciliary targeting of exogenous proteins, or the unique physicochemical property of primary cilia which is unfriendly to common immunostaining and imaging approaches [138,139,140,141]. Utilizing super-resolution microscopy, a recent study reported that PtdIns(3,4,5)P_3_ and PtdIns(4,5)P_2_ were visualized using validated antibodies in a ring shape at distinct subdomains of the inner transition zone membrane [53], indicating that more advanced microscopy imaging techniques are necessary for studying the sub-ciliary localization of PIs and PI-binding proteins.

In addition to identify more cilium-associated PIs and PI enzymes, it is vital to define more PI effectors. Many cilium-associated proteins are identified with PI-binding domains such as PH, C2, Tubby, and B9 domains [53,61,62,66,67,68,69,70,142,143]. Potential PI-binding motifs are also suggested in the cytoplasmic domains of ciliary transmembrane proteins, such as polycystins [144], via in vitro lipid binding assays. However, it has not been confirmed whether and to which PI species these proteins bind in vivo, and the consequent physiological significance of binding to these PIs remain unclear. In the future, it is vital to develop new tools, from novel biosensors and imaging methods to new mutations and animal models, to answer these questions. These studies will yield adequate knowledge to complete the route map of PI signaling in the context of primary cilia and development.

## Figures and Tables

**Figure 1 jdb-10-00051-f001:**
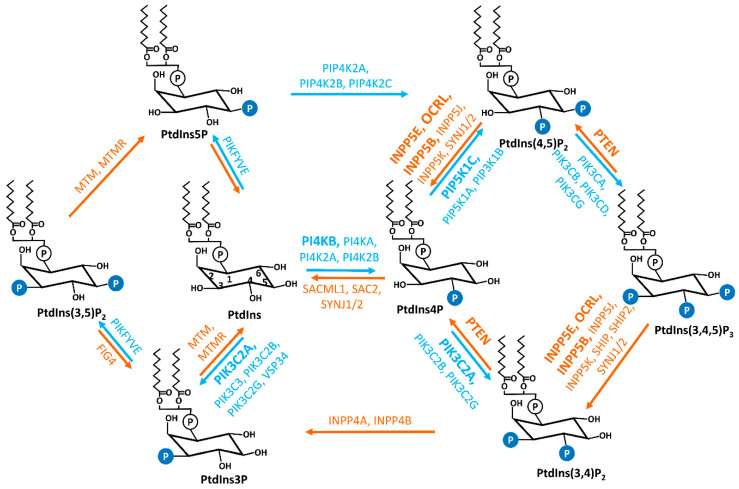
Interconversions of seven PIs. The reversible production of the PIs is mediated by the combined activities of specific phosphatases and kinases, which are labeled in orange and blue, respectively. The kinases and phosphatases that have been reported to be associated with cilia are marked in bold. PtdIns, phosphatidylinositol; PtdIns3P, phosphatidylinositol 3-phosphate; PtdIns4P, phosphatidylinositol 4-phosphate; PtdIns5P, phosphatidylinositol 5-phosphate; PtdIns(3,4)P_2_, phosphatidylinositol 3,4-phosphate; PtdIns(3,5)P_2_, phosphatidylinositol 3,5-phosphate; PtdIns(4,5)P_2_, phosphatidylinositol 4,5-phosphate; PtdIns(3,4,5)P_3_, phosphatidylinositol 3,4,5-trisphosphate; PI4KA, phosphatidylinositol 4-kinase alpha; PI4KB, phosphatidylinositol 4-kinase beta; PI4K2A, phosphatidylinositol 4-kinase type 2 alpha; PI4K2B, phosphatidylinositol 4-kinase type 2 beta; PIK3C3, phosphatidylinositol 3-kinase catalytic subunit type 3; PIK3C2A, phosphatidylinositol-4-phosphate 3-kinase catalytic subunit type 2 alpha; PIK3C2B, phosphatidylinositol-4-phosphate 3-kinase catalytic subunit type 2 beta; PIK3C2G, phosphatidylinositol-4-phosphate 3-kinase catalytic subunit type 2 gamma; PIP5K1A, phosphatidylinositol-4-phosphate 5-kinase type 1 alpha; PIP5K1B, phosphatidylinositol-4-phosphate 5-kinase type 1 beta; PIP5K1C, phosphatidylinositol-4-phosphate 5-kinase type 1 gamma; PIP4K2A, phosphatidylinositol-5-phosphate 4-kinase type 2 alpha; PIP4K2B, phosphatidylinositol-5-phosphate 4-kinase type 2 beta; PIP4K2C, phosphatidylinositol-5-phosphate 4-kinase type 2 gamma; PIKFYVE, phosphoinositide kinase, FYVE-type zinc finger containing; PIK3CA, phosphatidylinositol-4,5-bisphosphate 3-kinase catalytic subunit alpha; PIK3CB, phosphatidylinositol-4,5-bisphosphate 3-kinase catalytic subunit beta; PIK3CD, phosphatidylinositol-4,5-bisphosphate 3-kinase catalytic subunit delta; PIK3CG, phosphatidylinositol-4,5-bisphosphate 3-kinase catalytic subunit gamma; MTM, myotubularin; MTMR, myotubularin-related; SAC2, Sac domain-containing inositol phosphatase 2; SACML1, SAC1 like phosphatidylinositide phosphatase; SYNJ1/2, synaptojanin 1/2; INPP5E, inositol polyphosphate-5-phosphatase E; INPP5J, inositol polyphosphate-5-phosphatase J; OCRL, oculocerebrorenal syndrome of Lowe; INPP5B, inositol polyphosphate-5-phosphatase B; INPP5K, inositol polyphosphate-5-phosphatase K; PTEN, phosphatase and tensin homolog; INPP4A, inositol polyphosphate-4-phosphatase type I A; INPP4B, inositol polyphosphate 4-phosphatase type II; FIG4, factor-induced gene 4; SHIP, Src homology 2 (SH2) domain containing inositol polyphosphate 5-phosphatase.

**Figure 2 jdb-10-00051-f002:**
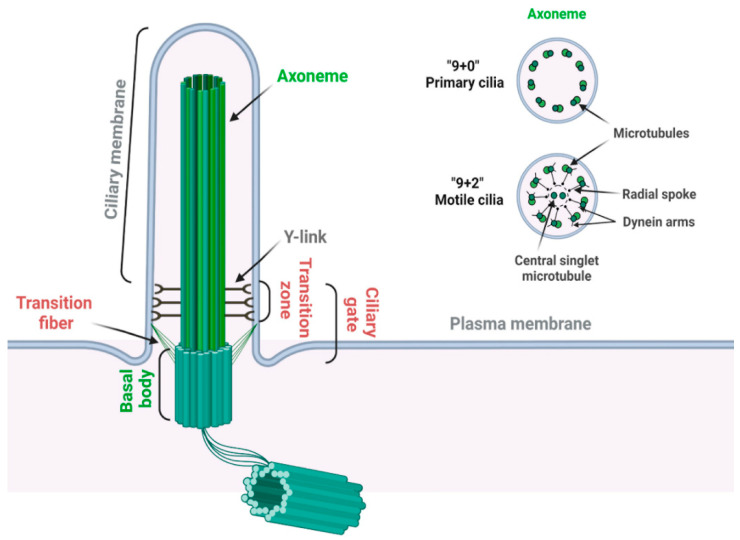
Structure of the primary cilium. A primary cilium is composed of a “9 + 0” microtubule axoneme and basal body complex (labeled in green). A motile cilium has two extra central singlet microtubules, generating a “9 + 2” arrangement (labeled in green). A motile cilium also contains dynein arms for ciliary movement and radial spoke to regulate the motility and motion pattern of motile cilia (labeled in black). Transition zone is characterized by the Y-links, which connect the proximal axoneme to the ciliary membrane. Transition fibers connect the distal end of the basal body to the ciliary membrane. Transition zone and transition fibers together coordinate the ciliary gate function (labeled in red) (created with BioRender.com, accessed on 18 November 2022).

**Figure 3 jdb-10-00051-f003:**
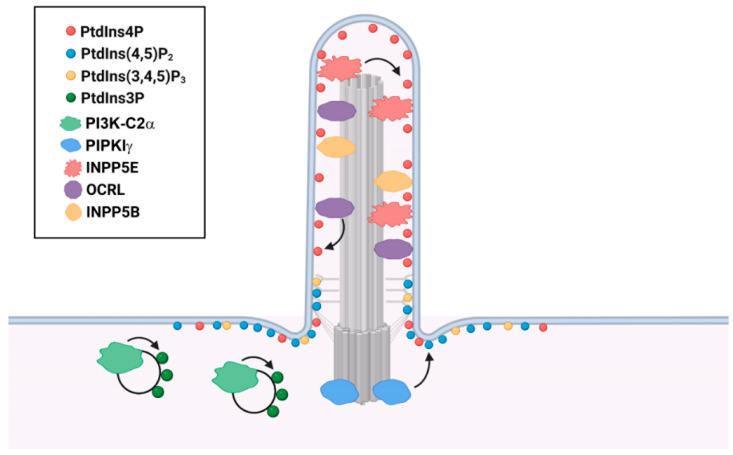
Ciliary localization of PIs and associated phosphatases and kinases. PtdIns4P is the main PI along the ciliary membrane. PtdIns(4,5)P_2_ and PtdIns(3,4,5)P_3_ mainly localize at the cilia base. PtdIns3P enriches in the pericentriolar recycling endocytic compartment at the ciliary base. The unique ciliary localization of PIs is generated by the specific phosphatases and kinases, as indicated by the arrows (Created with BioRender.com, accessed on 18 November 2022).

**Figure 4 jdb-10-00051-f004:**
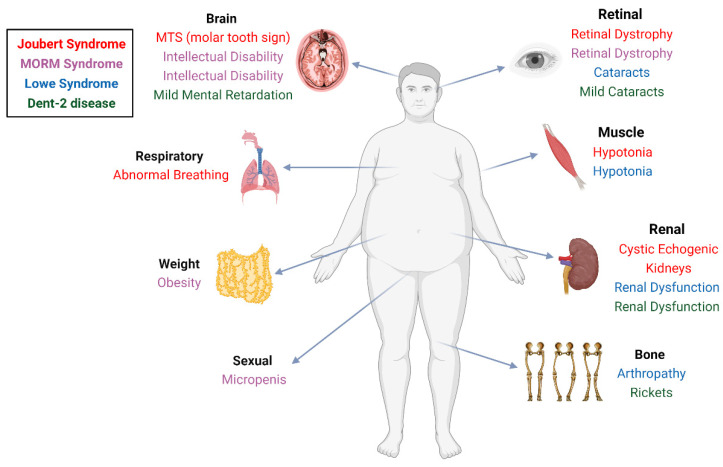
Clinical features in INPP5E and OCRL-associated ciliopathies or ciliopathy-like human diseases. Diseases and related symptoms are color-coded (Created with BioRender.com, accessed on 18 November 2022).

**Table 1 jdb-10-00051-t001:** Disease-causal mutations in INPP5E and OCRL.

Human Disease	Disease-Causal Mutations	Domain
Joubert Syndrome (INPP5E-associated)	R345S, R378C, T426N, R435Q, R435W, T436N, W474R, R512W, R515W, Y534D/R621Q, Y543X, G552A, R563H, K580E, V303M/R585C, Y588C	phosphatase domain
	G286R, R621Q/L234Pfs*56, C641R	Others ^1^
MORM Syndrome (INPP5E-associated)	Q627X, Q633X	Others ^1^
Lowe Syndrome (OCRL-associated)	F242S, W247C/X, W261C, Y272H, F276S, Q277R, Q295X, W297X, E302X, A328P, R334P/X, Y335X, R337C, G357E, R361I, T367del, V372G, N373Y, S374F, H414R, G421E, D422N, N424D, L448G, D451N/G, Q452R, R457G, K460X, F463S, E468K/G, P475H, R476W, Y477X, P478L, P495L, W497G/X, C498Y, D499H, R500G/Q/X, W503R, V508D, Y513C, S522R, H524R/Q, P526T/L, V527D, I533S	phosphatase domain
	V777E, E585del, N597K, L634P, F668V, C679W, L687P	ASH domain
	I768N, A797P, P801L, A861T, L891R	RhoGAP domain
	Q215X	Others ^1^
Dent-2 disease (OCRL-associated)	C87X, L56DfsX1, Q70RfsX18, E85FfsX26,	PH domain
	I257T, R301C, R301H, G304E, G321E, N354H, Y462C, R476W, P478L, Y479C, R493W	5-phosphatase
	E737D, P799L	RhoGAP domain
	T121NfsX1, I147KfsX1, S149X, P161PfsX3, M170IfsX1, F226S	Others ^1^
Lowe’s syndrome and Dent 2 disease (OCRL-associated)	I274T, F287S, R318C/H, D523N/G	5-phosphatase domain

^1^ The regions do not belong to any domains.

**Table 2 jdb-10-00051-t002:** Clinical features of INPP5E- and OCRL-associated human diseases.

Human Disease	Clinical Features
Joubert Syndrome (INPP5E-associated)	MTS (molar tooth sign), hypotonia, abnormal breathing, retinal dystrophy, cystic echogenic kidneys
MORM Syndrome (INPP5E-associated)	mental retardation, obesity, congenital retinal dystrophy, and micropenis in males
Lowe Syndrome (OCRL-associated)	mental retardation, congenital cataracts, maladaptive behavior, renal dysfunction, postnatal growth retardation, areflexia, and arthropathy
Dent-2 disease (OCRL-associated)	renal dysfunction, mild mental retardation, hypotonia, cataracts, and rickets

**Table 3 jdb-10-00051-t003:** Clinical features of PI3K-C2α and PIPKIγ-associated human diseases.

Human Disease	Disease-Causing Mutations
PI3K-C2α-associated	skeletal malformations, short stature, cataracts, secondary glaucoma, and developmental delay
LCCS3 (PIPKIγ-associated)	arthrogryposis with multiple joint contractures, muscle wasting, and atrophy

## Data Availability

Not applicable.

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
