# Peer review of "The Role of Primary Cilia-Associated Phosphoinositide Signaling in Development"

_jdb, 2022, doi:10.3390/jdb10040051_

Round 1

Reviewer 1 Report

In this review, the authors provide a detailed description of the importance of primary cilia-associated phosphoinositides in development and disease, in different processes regulating signaling pathways in primary cilia, including cilia stability and maintenance, and transport of components for axoneme elongation. It clearly emphasize the importance of spatiotemporal distribution of different phosphoinoside species in different compartments of the primary cilia in relevance to the function and the regulation by relevant phosphatases, kinases and PI binding effectors are described in detail. Although I am not an expert in this particular field to judge the scientific content much, I find this review in general well written, well organized and easily understandable. It would offer a nice overview for readers who are interested in this topic.

However here are my minor comments,

In lines 75 and 76, its mentioned that primary cilia is found on the most non-dividing cells. But its known to be present in both dividing and non-dividing cells, so this sentence could be rephrased.

Line 211: Numbering error, its 4.1 not 3.1

Line 297: Table 1 heading "Diseases-causal mutations in INPP5E and OCRL"

Line 299: Numbering error, its 4.2 not 3.2

Line 475 and 476: PI 3-phosphatase phosphatase and tensin homolog

Reviewer 2 Report

This review by Chen et al. focuses on the role of phosphoinositide lipids in cilia biology. The review is timely, well written and will be of interest to cilia biologists. I have some minor comments that can help improve the manuscript:

1.     Figure legend for Figure 1 should indicate what the different abbreviations mean. If kinases and phosphatases that mediate the interconversion between specific PI species remain unidentified, this should be mentioned in the legend as well.

2.     Lines 75-76: “non-dividing cells” can be misinterpreted as terminally differentiated cells, which is not the cell type authors are referring to here. Perhaps “quiescent cells” is more appropriate.

3.     Lines 78-82: A non-specialist will not be able to appreciate what a “central microtubule pair” means without more explanation. Please clarify or simplify this sentence. A schematic of a 9+0 axoneme would be helpful in Figure 2 if the authors want to mention this axonemal architecture for the primary cilium (Line 96).

4.     Line 136: Please change Gpr161 to GPR161.

5.     Line 171-172: “and facilitates the ciliogenesis” should be “and facilitates ciliogenesis”.

6.     Line 200: “worth to investigate” should be “worth investigating”.

7.     Line 206: “are unclear” should be “is unclear”.

8.     Figure 2: arrowheads should point to the structure in question, not the label.

9.     Figure 3: figure legend needs to expand on what the arrows mean. INPP5E is a membrane associated protein, but the schematic makes it look like it is axoneme-associated. Please check to make sure that the localizations (ciliary membrane vs. membrane associated vs. cilioplasm vs. axoneme) of OCRL and INPP5B are also properly represented in the schematic.

10.  Line 224: Should “PDE6d” be “PDE6D”?

11.  Line 231: please explain that a molar tooth sign indicates abnormal development of the cerebellum.

12.  Line 358: “just begins” should be “has just begun”.

13.  Line 365: Do authors mean “may play” instead of “may obsess”?

14.  Line 380: “are highly” should be “is highly”.

15.  Line 383: what does class II mean?
